# The G-Protein-Coupled Estrogen Receptor (GPER) Regulates Trimethylation of Histone H3 at Lysine 4 and Represses Migration and Proliferation of Ovarian Cancer Cells In Vitro

**DOI:** 10.3390/cells10030619

**Published:** 2021-03-11

**Authors:** Nan Han, Sabine Heublein, Udo Jeschke, Christina Kuhn, Anna Hester, Bastian Czogalla, Sven Mahner, Miriam Rottmann, Doris Mayr, Elisa Schmoeckel, Fabian Trillsch

**Affiliations:** 1Department of Obstetrics and Gynaecology, University Hospital, LMU Munich, Marchioninistr. 15, 81377 Munich, Germany; nanh14162@gmail.com (N.H.); sabine.heublein@med.uni-heidelberg.de (S.H.); christina.kuhn@uk-augsburg.de (C.K.); anna.hester@med.uni-muenchen.de (A.H.); bastian.czogalla@med.uni-muenchen.de (B.C.); sven.mahner@med.uni-muenchen.de (S.M.); 2Department of Obstetrics and Gynaecology, Heidelberg University Hospital, Im Neuenheimer Feld 440, 69120 Heidelberg, Germany; 3Department of Obstetrics and Gynaecology, University Hospital Augsburg, Stenglinstr. 2, 86156 Augsburg, Germany; 4Munich Cancer Registry (MCR), Bavarian Cancer Registry—Regional Center Munich (LGL), Institute for Medical Information Processing, Biometry and Epidemiology (IBE), Ludwig-Maximilians-University (LMU), 81377 Munich, Germany; rottmann@ibe.med.uni-muenchen.de; 5Department of Pathology, LMU Munich, Thalkirchner Str. 36, 80337 Munich, Germany; doris.mayr@med.uni-muenchen.de (D.M.); elisa.schmoeckel@med.uni-muenchen.de (E.S.)

**Keywords:** GPER, H3K4me3, GPER agonist G1, G15, ovarian cancer, cell migration and proliferation, p-ERK1/2

## Abstract

Histone H3 lysine 4 trimethylation (H3K4me3) is one of the most recognized epigenetic regulators of transcriptional activity representing, an epigenetic modification of Histone H3. Previous reports have suggested that the broad H3K4me3 domain can be considered as an epigenetic signature for tumor-suppressor genes in human cells. G-protein-coupled estrogen receptor (GPER), a new membrane-bound estrogen receptor, acts as an inhibitor on cell growth via epigenetic regulation in breast and ovarian cancer cells. This study was conducted to evaluate the relationship of GPER and H3K4me3 in ovarian cancer tissue samples as well as in two different cell lines (Caov3 and Caov4). Silencing of GPER by a specific siRNA and two selective regulators with agonistic (G1) and antagonistic (G15) activity were applied for consecutive in vitro studies to investigate their impacts on tumor cell growth and the changes in phosphorylated ERK1/2 (p-ERK1/2) and H3K4me3. We found a positive correlation between GPER and H3K4me3 expression in ovarian cancer patients. Patients overexpressing GPER as well as H3K4me3 had significantly improved overall survival. Increased H3K4me3 and p-ERK1/2 levels and attenuated cell proliferation and migration were observed in Caov3 and Caov4 cells via activation of GPER by G1. Conversely, antagonizing GPER activity by G15 resulted in opposite effects in the Caov4 cell line. In conclusion, interaction of GPER and H3K4me3 appears to be of prognostic significance for ovarian cancer patients. The results of the in vitro analyses confirm the biological rationale for their interplay and identify GPER agonists, such as G1, as a potential therapeutic approach for future investigations.

## 1. Introduction

Epithelial ovarian cancer (EOC) is the fifth leading cause of female cancer-associated death in Western countries [1]. One of the main reasons for the high mortality is delayed diagnosis in the advanced stage, when the cancer is already disseminated within the abdomen [2]. Given the heterogeneity of ovarian cancer, novel molecular drug targets need to be identified to tailor innovative personalized treatment approaches [3].

Estradiol (E2) is an important determinant of gynecologic malignancies, including ovarian cancer [4,5]. In this context, G-protein-coupled estrogen receptor (GPER) is a new member of the G-protein-coupled receptor (GPCR) family, mediating signals into the cells via its trans-membrane domains [6]. In 2000, a study reported that rapid 17β-estradiol-mediated activation of extracellular signal-regulated kinases (ERKs) was dependent on the protein of an orphan G-protein-coupled receptor with seven transmembrane domains [7]. In 2007, it was named as G-protein-coupled estrogen receptor 1 (GPER) [6]. GPER can initiate many early non-genomic signaling events of estrogen, such as enrichment of cAMP production, intracellular calcium mobilization, transactivation of EGFR and activation of PI3K/Akt as well as the MAPK pathway [8,9,10]. In GPER- and ERα-positive ovarian cancer cells, EGFR/ERK signaling is activated by 17β-estradiol and the selective GPER ligand G1, respectively; in contrast, however, the progesterone receptor only responds to E2 [11]. It has suggested that a functional interaction between GPER and ERα may exist in tumor cells [12].

Previous studies have demonstrated that activation of GPER inhibits proliferation and migration in different human cancer cell lines [13,14,15]. Activation of GPER could inhibit the growth of colorectal cancer cells in vivo via sustained ERK1/2 activation [15]. A similar result has been obtained in prostate cancer PC-3 cells, in that activation of GPER maintains phosphorylation of the ERK1/2 level, resulting in the arrest of PC-3 growth [14]. However, controversial results have been reported regarding the role of GPER in ovarian carcinogenesis [16,17,18]. Previous findings established that GPER stimulated the proliferation, migration and invasion in a ligand-independent manner in ovarian cancer SKOV3 cells [17]. Ignatov et al. elucidated that activation of GPER by G1 (1-[4-(6-bromobenzo [1,3]dioxol-5-yl)-3a,4,5,9b-tetrahydro-3*H*-cyclopenta[c]quinolin-8-yl]-ethanone), the selective synthetic agonist of GPER, suppressed proliferation of SKOV3 and OVCAR3 cells by inducing cell apoptosis and partially cell cycle arrest and was associated with increased phosphorylation of H3 [19]. The results imply that activated GPER may alter histone modifications in ovarian cancer. Moreover, the GPER selective antagonist G15, with a similar structure to G1, was identified in 2009 [20]. G15 is effective in inhibiting all G1-induced effects tested to date and many 17β-estradiol-mediated effects [6].

In contrast to the phosphorylation, histone H3 trimethylation at lysine 4 (H3K4me3) is a marker for trimethylation at the 4th lysine residue of the histone H3 protein, representing one of the best studied post-translational histone modifications [21]. The H3K4me3 modification is strongly associated with transcriptional initiation and elongation [22]. Different H3K4me3 profiles between cell lines link with transcriptional differences between the cell lines [23]. H3K4me3 is widely recognized as an active promoter that is positively correlated with gene expression [24] and closely involved in the tumor suppressor genes, treatment and prognostic outcome of cancer patients [25,26,27]. It has been demonstrated that NEK2 promotes proliferation, migration and tumor growth of gastric cancer cells through regulating the level of H3K4me3 [28]. Another study has demonstrated that broad H3K4me3 domains catalyzed by MLL4 (a COMPASS-like enzyme) are linked to transcriptional activation by interacting with super-enhancers at the tumor-suppressor genes in brain cancer cells [29]. Our own data suggests that 1α, 25(OH)_2_D_3_ (calcitriol) is able to induce H3K4me3 expression via vitamin D receptor in ovarian cancer [30]. In addition, an association of H3K4me3 with estrogen receptor α (ERα)-regulated gene transcription has been described [31]. Accumulation of H3K4me3 during ERα-activated transcription has been noted [32], while further studies point out that H3K4me3 expression is increased by ERβ stimulation [33]. According to the existing evidence, GPER is considered as a non-classical estrogen receptor [6], which is associated with 17β-estradiol-mediated rapid signaling events and transcriptional regulation [6,12]. However, the interplay of GPER and H3K4me3 in cancer biology has not been systematically investigated so far.

To study the relationship of GPER and H3K4me3 in ovarian cancer, we examined the expression levels of H3K4me3 in ovarian cancer specimens by immunohistochemistry, and followed these results in vitro by thoroughly investigating the effect of GPER in ovarian cancer cells via its specific agonist G1, antagonist G15 and knockdown of GPER expression.

## 2. Materials and Methods

### 2.1. Ethics Approval

All epithelial ovarian cancer specimens were derived from the archives of the Department Gynaecology and Obstetrics in LMU Munich, which were initially applied for pathological diagnostics. In all cases, the diagnostic procedures were completed before the current study was performed. Our study was approved by the Ethics Committee of the Ludwig-Maximilians-University (Date: 30 September 2009; approval number: 227-09; Munich, Germany). All experiments in this study were conducted in accordance with the Declaration of Helsinki. The authors were blinded to the patient data throughout the experimental analysis.

### 2.2. Patients and Tissue Microarray

The tissue microarray was conducted with 156 EOC tissue specimens obtained from patients who underwent surgery for EOC in the Department of Obstetrics and Gynecology of the Ludwig-Maximilians-University Munich between 1990 and 2002. Clinical data was derived from patient charts and follow-up data were obtained from the Munich Cancer Registry. All samples were prepared by formalin fixation and paraffin embedding (FFPE). Three representative tissues were taken from each patient for the microarray analysis to obtain a more accurate image of EOC [34].

### 2.3. Immunohistochemistry

The paraffin-embedded and formalin-fixed samples from 156 EOC patients were used to construct a tissue microarray (TMA). Sections of 3 μm were cut from the TMA block and immunohistochemical (IHC) staining for GPER and H3K4me3 was performed as previously described [30,35]. The intensity of the expression was evaluated by the immunoreactive score (IRS).

### 2.4. Chemicals and Cell Culture

G1 (selective GPER agonist, Catalog No.: 3577) and G15 (selective GPER antagonist, Catalog No.: 3678) were purchased from Tocris. Both of them were dissolved with DMSO and stored at −20 °C according to company protocol.

The human ovarian cancer cell lines Caov3 and Caov-4 were purchased from the American Tissue Culture Collection (ATCC, Wesel, Germany). While Caov-4 is derived from a metastatic site of a 45-year-old female with high-grade serous histology, Caov3 is from the primary tumor of a 54-year-old Caucasian female with adenocarcinoma of the ovary. Both cell lines were routinely cultured in RPMI1640 (ThermoFisher Scientific, Waltham, MA, USA) supplemented with 10% fetal bovine serum (FBS) at 37 °C in a humidified 5% CO2 air. The medium was changed into phenol red-free RPMI1640 (ThermoFisher Scientific) plus 10% charcoal stripped FBS (Gibco, New York, NY, USA) 24 h before all experiments. Cells were treated with 0.1% (*v*/*v*) dimethyl sulfoxide (DMSO) as the vehicle.

### 2.5. MTT Assay

Cell viability was assessed by doing a 3-(4, 5-dimethylthiazol-2-yl)-2, 5-diphenyltetrazolium-bromide assay (MTT assay). Cells were seeded into 96-well plates at a density of 5.0 × 10^3^/well in triplicates and incubated for 24 h. After this, cells were treated by a concentration gradient of G1 for 24 h in the incubator, followed by the addition of 20 µL of 5 mg/mL MTT (Sigma-Aldrich Co., St. Louis, MO, USA) to each well for 90 min at 37 °C. The culture medium with MTT was removed. Purple formazan crystals were dissolved in 200 μL DMSO/well and then mixed thoroughly on the shaker for 5 min at room temperature. The optical density (OD) was read at 595 nm using an Elx800 universal Microplate Reader. The experiments were repeated three times at least. The IC50 values were calculated by GraphPad Prism 8 software.

### 2.6. siRNA Transfection

Caov3 and Caov4 cells were seeded into 6-well plates until 60–80% confluence before transfection. GPER-specific siRNA (QIAGEN Sciences, Cat.No.1027416, Germantown, MD, USA) or non-specific control siRNA (QIAGEN, Cat.No.1027280) were transiently transfected using Lipofectamin RNAiMAX reagent (Invitrogen, Carlsbad, CA, USA) according to the manufacturer‘s instructions. The level of GPER protein in transfected cells was analyzed by Western blotting to verify the effect of transfection after 48 h.

### 2.7. Preparation of Cell Lysates and Western Blot

Western blot analysis was performed with the cell lysates of different groups of treated cells or untreated cells. Then cells were harvested and lysed by RIPA buffer (Sigma-Aldrich, Steinheim, Germany). The protein concentration of the supernatant was determined by Bradford assay. Proteins were separated by electrophoresis in 10% or 12% SDS-PAGE gels and transferred to PVDF membranes. The membranes were incubated overnight at −4 °C with a 1:2000 dilution of GPER antibody (LS-C203019, Lifespan Biosciences, Seattle, WA, USA), 1:1000 dilution of ERα antibody (ab79413, Abcam, Cambridge, UK), 1:200 dilution of ERβ antibody (53472, Anaspec, Fremont, CA, USA), 1 μg/mL dilution H3K4me3 antibody (ab8580, Abcam), 1:2000 dilution of Histone 3 antibody (#4499, Cell Signaling Technology, Danvers, MA, USA), 1:500 dilution of p-ERK1/2 (ab47339, Abcam) or 1:1000 dilution of β-actin antibody (A5441, Sigma, Kawasaki, Japan). Since the Caov3 cell line has been previously published to produce GPER by our institution [35], it was also used as a positive control of GPER in our study. In addition, we utilized the same antibody of GPER as our previous studies [35,36,37,38]. Afterwards, the membranes were washed three times and incubated for 2 h with a 1:1000 dilution of the corresponding alkaline phosphatase-conjugated secondary antibodies. Blotting was detected and visualized by BCIP/NBT Color Development Substrate (Promega, Madison, WI, USA).

Images were analyzed by an image analyzer (Molecular Imager^®^ Gel DocTM XR+, Bio-rad) using software Quantity One 4.6.7 (Bio-Rad, Munich, Germany). β-actin was used as an endogenous control. Results shown are representative of at least three biological replicates.

### 2.8. Wound Healing (Scratch) Assay

The wound healing assay was used to evaluate the cell migration of two ovarian cancer cell lines upon treatment. The cells (1 × 10^6^/well) and siRNA-transfected cells were seeded into 24-well plates, allowed to form a confluent monolayer and serum-starved overnight. A scratch was made by a sterile 200 µL pipette tip in order to create a denuded gap of constant width. Each well was rinsed once with PBS to remove the detached cells. Cells were exposed to different reagents (serum-free culture medium), including the control with the vehicle,1 μM G1, 1 μM G15, 1 μM G1 with 1 μM G15, GPER siRNA with vehicle and GPER siRNA with 1 μM G1. Cells were incubated at 37 °C in the presence of 5% CO_2_. The migrations of the cell lines were monitored at different time points (0 h and 48 h), using an inverse phase-contrast microscope (Leica Dmi1, Leica, Wetzlar, Germany) with a camera (LEICA MC120 HD, Leica, Wetzlar, Germany). The experiments were repeated three times. Finally, an image of the wound area was measured by Image J software. The migration of cells toward the wound was expressed as a percentage of wound closure:The percentage (%) wound closure = ((At = 0 h − At = Δ h)/At = 0 h) × 100%(1)
In our study, At = 0 h means the area of wound measured immediately after a scratch, and At = Δ h means the area of wound measured 48 h after a scratch.

### 2.9. BrdU Assay

Cell proliferation was determined by BrdU assay (Roche Applied Science, Mannheim, Germany) based on the measurement of a pyrimidine analogue (BrdU) incorporation during DNA synthesis. The experiments were performed in accordance with the manufacturer’s instruction. Briefly, the Caov3 and Caov-4 cell lines were planted in triplicate in 96-well plates (5 × 10^3^ cells/well) and incubated for 24 h in the incubator. To detect the effect of the GPER on the Caov3 and Caov4 cells, GPER-knockdown cells (5 × 10^3^ cells/well) were seeded in triplicate in 96-well plates simultaneously. Cells were treated with different reagents, including the control with the vehicle, 1 μM G1, 1 μM G15, 1 μM G1 with 1 μM G15, GPER siRNA with vehicle and GPER siRNA with 1 μM G1. After 24-h of treatment, the cells were labelled with BrdU and incubated for 2 h at 37 °C. Anti-BrdU-POD (100 µL/well) was added and remained for 90 min at room temperature after cell fixation, followed by washing three times. A total of 8 min after the substrate solution was added to each well, the reaction was stopped by adding 1M H_2_SO_4_ (25 μL/well). The absorbance of the samples at 450 nm was measured using an Elx800 universal Microplate Reader. All experiments were repeated in triplicate.

### 2.10. Statistical Analysis

A Spearman rank test was performed for correlations between the continuous variables. Survival times were compared using the Kaplan–Meier (log-rank) test method. The ROC curve and Youdan index were used to identify an appropriate cut-off [39,40]. All values in vitro were reported as the mean ± SEM of three independent experiments. Data were analyzed by a two-tailed student’s *t*-test between two groups and one-way ANOVA in multiple groups. The statistical analyses were performed using SPSS version 25.0 (IBM, Armonk, NY, USA). A *p*-value of <0.05 was considered to be statistically significant for all analyses.

## 3. Results

### 3.1. H3K4me3 Correlates with GPER Expression in EOC

The combination of both GPER and H3K4me3 stainings was technically feasible in 146 of 156 cases (93.6%), including serous (103 of 146 cases, 70%), clear cell (11 of 146 cases, 8%), endometrioid (20 of 146 cases, 14%) and mucinous carcinomas (12 of 146 cases, 8%). Of this cohort, missing cases due to technical failure accounted for 6.4% (10 of 156). According to previous studies of GPER and H3K4me3 localization, GPER staining was predominantly observed in the cytoplasm and membrane in ovarian cancer specimens, while H3K4me3 staining primarily localized within the nuclei of EOC cells (Figure 1). Detailed information of the GPER and H3K4me3 immunoreactive score (IRS) regarding clinical and pathological characteristics have already been reported in recently published data by our group [30,35]. Analyses show that a nuclear H3K4me3 expression is positively correlated with GPER expression (Rho = 0.177; *p* = 0.033). Representative staining of GPER and H3K4me3, from the same patient, is shown in Figure 1.

### 3.2. H3K4me3 Expression Has Prognostic Impact in GPER Positive EOC Patients

In our previous work, we demonstrated no significant difference in prognosis for EOC patients whether whose tumors expressed GPER or not [35]. To examine the clinical significance of tumors expressing both H3K4me3 and GPER, survival analyses were conducted by the Kaplan–Meier curves with a log-rank test. When cases with higher expression of GPER (IRS = 6–12) were evaluated, strong expression of H3K4me3 (IRS = 9–12) was associated with a favorable prognosis (median overall survival not reached vs. 39.0 months, *p* = 0.037, Figure 2).

### 3.3. Expressions of GPER, ERα and ERβ in Caov3 and Caov4 Cell Lines

First of all, we detected the expressions of GPER, ERα and ERβ in the Caov3 and Caov4 cell lines by Western blot (Figure 3). MCF-7 breast cancer cells were used as a positive control of ERα and ERβ. GPER expression was observed in both of the Caov3 and Caov4 cell lines. Compared with the Caov3 cell line, Caov4 cells expressed a stronger GPER protein (Figure 3C, ** *p* < 0.01). Our result showed that the Caov4 cells expressed strongly the ERα protein; however, the expression of ERα was negative in Caov3 cells (Figure 3B,D). In addition, ERβ was not expressed in either of the Caov3 and Caov4 cell lines (Figure 3B).

To confirm that GPER was involved in the G1-induced proliferation and migration of ovarian cancer cells, and in relationship with other protein expressions, we knocked down GPER expression with GPER siRNA. Western blot results showed obvious decreases in GPER expression in the GPER siRNA-transfected groups compared with that of the control groups (Figure 4, ** *p* < 0.01). An increasing level of GPER expression and the decreasing level of GPER expression were found, respectively, after 1 μM G1 and 1 μM G15 treatment for 24 h in the Caov3 and Caov4 cell lines (Appendix A).

### 3.4. IC50 of G1 in Ovarian Cancer Cells

To investigate whether GPER could influence the growth of ovarian cancer cells, Caov3 and Caov4 cells were incubated with increasing concentrations of the GPER-specific agonist G1 for 24 h. A concentration-dependent inhibition of cell viability was observed (Figure 5). The estimated IC50 value was 2.25 μM and 2.29 μM for Caov3 and Caov4, respectively. In addition, we also extended our experiments using G36, a newly discovered GPER antagonist, to treat Caov3 and Caov4 cells for 24 h. The results showed that only a very high concentration of G36 could suppress cell viability in Caov3; however, G36 was not able to influence the growth of Caov4 cells (Appendix A). Therefore, 1 μM was used as an exposure concentration of G1 and G15 (a selective GPER antagonist) to carry out the next experiments of our research.

### 3.5. The Selective GPER Agonist G1 Inhibits Cell Proliferation via Activation of GPER in Caov3 and Caov4 Cell Lines

To explore the function of GPER in ovarian cancer cells, we administered 1 μM G1, 1 μM G15 and 1 μM G1 with 1 μM G15 to ovarian cancer cells and the treatment time was 24 h. Then cell proliferation was detected by using a BrdU assay. Obvious suppression of cell proliferation was observed in Caov3 and Caov4 cell lines after 1 μM G1 treatment for 24 h (Figure 6, *** *p* < 0.001 vs. control group). Additionally, 1 μM G15 treatment for 24 h was able to increase cell proliferation of Caov4 cells (Figure 6B, ** *p* < 0.01 G15 group vs. control group). G15 had no effect on the Caov3 cell line alone (Figure 6A, *p* > 0.05). Our findings also suggested that G15 played an opposing role on proliferation of Caov4 cells as compared to G1 treatment (Figure 6B, ^###^
*p* < 0.001 G1 + G15 group vs. G1 group; ^$^
*p* < 0.05, G15 group vs. G1 + G15 group). In the Caov3 cell line, we did not obtain a significant difference between the G15 group and G1 + G15 group; however, remarkable differences were found between the G1 group and G1 + G15 group (Figure 6A, ^##^
*p* < 0.01, G1 group vs. G1 + G15 group). All the results suggested that G15, the selective GPER antagonist, could block the inhibitory action of G1 to cell proliferation in the Caov3 and Caov4 cell lines.

To further explore the involvement of GPER activation in the inhibitory effect of G1 on ovarian cancer cells, we extended our experiments by knockdown of GPER and stimulation of G1 together. We knocked-down GPER with GPER siRNA and then treated the GPER-knockdown cells with 1 μM G1 for 24 h. There were no statistical discrepancies between the control groups and GPER siRNA-transfected groups in Caov3 and Caov4 cells (Figure 6, *p* > 0.05). It indicated that the GPER-knockdown of Caov3 and Caov4 cells could not inhibit cell proliferation. Significant differences were observed between the G1 groups and GPER siRNA with the G1 groups in both Caov3 and Caov4 cell lines (Figure 6, ^##^
*p* < 0.01 and ^###^
*p* < 0.001 vs. G1 group, respectively). In addition, there were no differences between the GPER siRNA groups and GPER siRNA + G1 groups (Figure 6A,B, *p* > 0.05). These results showed that G1 was not able to exert its inhibitory action onto the GPER-knockdown Caov3 and Caov4 cells. Taken together, our results suggest that GPER activation is involved in the G1-induced inhibitory effect on Caov3 and Caov4 cells.

### 3.6. G1 Suppresses Cell Migration through Activation of GPER in Caov3 and Caov4 Cell Lines

We observed the effects of activated GPER by G1 on cell migration of ovarian cancer cells by the wound healing assay. The M30 apoptosis assay was first carried out to verify that G1-induced alteration of cell migration was not because G1 caused significant cell death. There were no differences found in the Caov3 and Caov4 cell lines (Appendix A). Our data of the wound healing assay demonstrated that 48 h treatment with 1 μM G1 decreased the migration intensity of the Caov3 and Caov4 cells considerably (Figure 7). To confirm that GPER took part in the regulation of cell migration caused by G1, we knocked down the GPER protein with GPER siRNA and then treated the GPER-siRNA cells with 1 μM G1 for 48 h. Knocking down the GPER protein of Caov3 and Caov4 cells could not influence their cell migration (Figure 7, *p* > 0.05 vs. control group). Our data displayed that the percentage of wound closure of GPER siRNA with G1 treatment groups were significantly higher than that of the G1 groups in Caov3 (Figure 7A,C, ^##^
*p* < 0.01 G1 group vs. GPER siRNA + G1 group) and Caov4 (Figure 7B,D, ^##^
*p* < 0.01 G1 group vs. GPER siRNA + G1 group) cells. Moreover, no obvious differences between the GPER siRNA groups and GPER + G1 groups were found in the Caov3 and Caov4 cell lines (Figure 7, *p* > 0.05). Our data indicated that 1 μM G1 led to the inhibition of cell migration via the activation of GPER in Caov3 and Caov4 cells.

We extended our research by using 1 μM G15 to block the action of G1. No notable differences were found between the control group and G15 group in the Caov3 cell line (Figure 7A,C, *p* > 0.05). In addition, G15 was not able to block the action of G1 in Caov3 cells (*p* > 0.05, G1 group vs. G1 + G15 group). However, 1 μM G15 caused a great increased capacity of cell migration in the Caov4 cell line (Figure 7B,D, * *p* < 0.05). Additionally, 1 μM G15 was capable of counteracting the action of G1 in cell migration in Caov4 (Figure 7B,D, ^##^
*p* < 0.01 G1 group vs. G1 + G15 group and ^$$^
*p* < 0.01 G15 group vs. G1 + G15 group).

### 3.7. G1 Treatment Elevatesthe Levels of ERK1/2 Phosphorylation and H3K4me3 via Activation of GPER

A previous study has reported that phosphorylated ERK1/2 is frequently a common endpoint of activation of GPER [41]. Therefore, we investigated the levels of phosphorylated ERK 1/2(p-ERK1/2) and H3K4me3 after G1 treatment for the indicated times in order to confirm the GPER pathway was activated by G1. In this study, we found that G1 treatment (30–60 min) was able to rapidly increase the phosphorylation of the ERK1/2 and H3K4me3 levels (Figure 8). The phosphorylation of ERK1/2 and H3K4me3 were maintained for 24 h in both the Caov3 and Caov4 cells (Figure 8).

To confirm GPER was involved in regulation of p-ERK1/2 and H3K4me3, the GPER-specific antagonist G15, G1 + G15, siRNA-specific targeting to GPER and GPER siRNA with G1 were utilized. The levels of p-ERK1/2 and H3K4me3 proteins were investigated by Western blot assay. As shown in Figure 9 and Figure 10, G1 treatment activated p-ERK1/2, as seen by the increased levels of the phosphorylation status. Meanwhile, increasing levels of H3K4me3 protein were found after G1 treatment for 24 h in both cell lines. Knockdown of GPER significantly attenuated phosphorylated ERK1/2, as shown by a significant decrease (Figure 9 and Figure 10, * *p* < 0.05 or ** *p* < 0.01, GPER siRNA group vs. control group). No remarkable differences of p-ERK1/2 and H3K4me3 levels were found between the GPER siRNA and GPER siRNA+G1 group of Caov3 (Figure 9, *p* > 0.05) and Caov4 cells (Figure 10, *p* > 0.05). Furthermore, 1 μM G1 treatment for 24 h was not able to enhance phosphorylated ERK1/2 and H3K4me3 protein of GPER-knockdown Caov3 and Caov4 cells (Figure 9 and Figure 10, ^##^
*p* < 0.01 or ^###^
*p* < 0.001 G1 group vs. GPER siRNA + G1 group). All the results suggested that G1 treatment activated ERK1/2 rapidly and maintained the phosphorylated status for 24 h. At the same time, activation of GPER by G1 elevated the levels of p-ERK1/2 and H3K4me3 in Caov3 and Caov4 cells.

We also used 1 μM G15 and 1 μM G1 + 1 μM G15 to treat ovarian cancer cells for 24 h, respectively, to extend our study. Obviously reduced levels of p-ERK1/2 and H3K4me3 were observed after G15 treatment in Caov4 cells (Figure 10, * *p* < 0.05 or *** *p* < 0.001 vs. control group). G15 had no effect on regulation of p-ERK1/2 and H3K4me3 proteins in the Caov3 cell line. In addition, G15 blocked the enhancement of the p-ERK1/2 and H3K4me4 levels in Caov3 cells (Figure 9A–C) and Caov4 cells (Figure 10A–C). The alteration of H3K4me3 staining was displayed after G1 or G15 treatment for 24 h in Caov3 and Caov4 cells (Appendix A).

## 4. Discussion

In our study, we elucidated a positive correlation between GPER and H3K4me3 in EOC patients. High expression of both GPER and H3K4me3 was associated with a favorable prognosis. Using in vitro experimental models, we demonstrated that the G1 treatment significantly inhibited the proliferation and migration of Caov3 and Caov4 cells, and increased the levels of p-ERK1/2 and H3K4me3 proteins via activation of GPER (Figure 11). Conversely, GPER antagonist G15 was able to block the inhibitory effect of G1 on cell proliferation and migration and induce the attenuation of H3K4me3 protein in the Caov4 cell line. We therefore proposed that activated GPER induced H3K4me3 expression and therapeutic approaches addressing this interplay might have the potential to reduce of migration and growth of ovarian cancer cells impacting the clinical behavior of the disease.

At present there have been several reports about GPER and clinical outcomes of ovarian cancer patients, providing very confusing and controversial results. Smith and co-workers found that the GPER expression was linked to lower survival rates in ovarian cancer [42]. However, the contrary result was observed in another study, which suggested a loss of GPER expression during ovarian tumorigenesis and that GPER expression was associated with a favorable clinical outcome [19]. A previous study in our institution reported that, in EOC tissue, GPER was related to prolonged overall survival in FSHR (follicle stimulating hormone receptor)/LHCGR (luteinizing hormone receptor)-negative patients [35]. Trimethylation of histone H3 lysine 4 (H3K4me3) is a well-known modification linked to gene transcription and enriched in gene promoters [22]. The prognostic relevance of H3K4me3 modification has been reported by others in different human cancers. Recent studies demonstrated that the increased expression of H3K4me3 was associated with an improved prognosis in bladder and renal cell carcinomas [43,44]. The opposite results were also reported, in that a higher level of H3K4me3 was correlated with a worse prognosis in hepatocellular cancer [45] and breast cancer [46]. In our study, our data revealed the first time that GPER was positively related with H3K4me3 expression and a higher level of H3K4me3 in combination with GPER was associated with a better prognosis outcome. Our result was partly consistent with the previous results. These discrepancies might be explained by different patterns of GPER and H3K4me3 in different types of cancers and in various histologic types of some cancers. The heterogenicity of H3K4me3 was also proved in different cell lines, as colorectal cancer cells expressed obviously lower levels of H3K4me3 than normal cells [47], whereas gain of H3K4me3 was remarkably collected with late-stage breast cancer cells [46]. A previous study showed that the low level of H3K4me3 was associated with adverse clinical-pathological parameters and correlated with patients’ outcome in renal cell carcinoma [44]. Moreover, a recent study suggested that broad H3K4me3 was able to result in activation of tumor suppressor genes and repression of oncogenes [25]. Therefore, we assumed that activation of GPER might increase the level of H3K4me3 and, in the meanwhile, suppress the growth of ovarian cancer cells.

In agreement with the expression analyses, controversial results regarding the role of GPER were shown in different cancer cell lines. Our results in demonstrating the inhibitory effects of G1 on cell proliferation and migration of Caov3 and Caov4 cells (Figure 5) are consistent with the observations released in other ovarian cancer models; however, they are inconsistent with other studies that reported GPER promoted growth of the SKOV3 [17] and OVCAR5 [18] cell lines. The contradictory results might be due to the utilization of agonists or antagonist of different specificities, such as estrogen and 1,3-Bis(4-hydroxyphenyl)-4-methyl-5-[4-(2-piperidinylethoxy)phenol]-1*H*-pyrazole dihydrochloride (MPP), in their studies, as compared to the use of the specific agonist G1 and antagonist G15 of GPER in our study. Interestingly, a recent study reported that G1 treatment suppressed proliferation and induced apoptosis of KGN cells (a human ovarian granulosa cell tumor cell line) in a GPER-independent manner [16]. This surprised observation was probably explained that tremendous heterogeneity was exhibited among different ovarian cancer cell lines [48,49]. Therefore, alteration of GPER expression was detected after G1 and G15 treatment for 24 h in our study. As Appendix A shown, the level of GPER was increased by G1 treatment, and G15 treatment decreased GPER expression in Caov3 and Caov4 cells.

GPER (G-protein-coupled estrogen receptor) was discovered as a member of the G-protein-coupled receptor (GPCR) family [50]. The activation of GPER could regulate the activities of multiple downstream signals, such as PI3K/Akt and EGFR/ERK signaling [10]. A previous study suggested that ERK1/2 was a reliable marker of agonist-mediated GPCR and used to measure the functional outcome of receptor stimulation [51]. Additionally, regardless of the signaling pathway, phosphorylated ERK1/2 was typically regarded as a common endpoint of activation for GPER [41]. Our data revealed that G1 treatment rapidly increased p-ERK1/2 and could sustain the p-ERK1/2 level for 24 h. Similar results were also observed in breast [52], prostate [14] and adrenocortical cancer cells [53]. Furthermore, the level of H3K4me3 was promoted by G1 treatment in a time-dependent manner. All the results suggest that, in our study, GPER and its downstream targets were indeed activated by G1 treatment for 24 h.

A previous study by others suggested that a functional interaction between GPER and ERα might occur when the tumor cells expressed both of the receptors [12]. As Caov4 expressed ERα as well as GPER (Figure 3), it was difficult to pinpoint which of both receptors was responsible for regulating the signaling pathway and growth-inhibitory action. To investigate the specific effect of GPER in ovarian cancer cells. We used G1 (a selective ligand for GPER) in our study. Although G36 was seen a more selective GPER antagonist than G15 [54], our data showed only suppression of cell proliferation was found for a very high concentration of G36-treated Caov3 cells (Appendix A). G15 has a similar structure to G1 [20], and is effective in inhibiting all G1-mediated effects [20,55,56]. Therefore, G15 was used to block GPER and the actions of the G1 treatment in our experiments. G1 and G15 both have no affinity for ERα [20,57]. According to the estimated IC50 of G1 for the Caov3 and Caov4 cell lines, we used 1 μM G1, 1 μM G15 and 1 μM G1 + 1 μM G15 to treat cells. Our results suggested that inhibition of the proliferation and migration of Caov3 and Caov4 cells was attributable primarily to activation of GPER by G1 treatment. This hypothesis was confirmed by siRNA knockdown of GPER, which blocked the G1 inhibitory effect on cell growth of Caov3 and Caov4 cells. Several recent investigations suggested that activation of GPER by high concentration (1 μM) G1 treatment played an inhibitory role in various cancer cells [15,19,53], which are in agreement with our findings. Furthermore, 24 h treatment with G15 reversed the inhibitory effect of G1 and blocked the actions of G1 in our extended experiments. Our study provided evidence that G1 treatment suppressed proliferation and migration of Caov3 and Caov4 cells via activation of GPER.

The mode of action of G1 and its interaction with GPER in Caov3 and Caov4 cells have not been studied previously. In the ERα-negative breast cancer cells (MDA-MB-468 and MDA-MB-436 cells), rapid activation of ERK1/2 (<30min) was induced by G1 treatment via GPER/EGFR/ERK signaling, leading to cell proliferation and cell migration [58]. On the contrary, G1 treatment was shown to induce sustained ERK1/2 in Caov3 and Caov4 cells, but the biological consequence was a profound inhibition of cell proliferation and migration in our study. Our results are supported by previous findings, which showed that p-ERK1/2 was maintained for 24 h and cell growth was suppressed under the treatment of G1 in breast [52], prostate [14] and adrenocortical cancer cells [53]. Both ERK1/2 activation and cell growth suppression were dependent on GPER, as siRNA knockdown and the antagonist of GPER effectively blocked the actions of G1.

It has been demonstrated that ERK1/2 is part of the GPER-mediated pathway [7]. The activation of ERK1/2 plays a central role in cell proliferation control [59]. Activation of ERK1/2 (p-ERK1/2) is able to translocate from the cytoplasm to the nucleus to phosphorylate their nuclear targets for transcriptional regulation [60]. A previous study has suggested cytosolic ERK1/2 activation inhibits survival and proliferation signals in the nucleus [60]. Moreover, the duration of ERK1/2 phosphorylation can determine a cell’s fate, with transient p-ERK1/2 resulting in cell survival and proliferation and prolonged p-ERK1/2 with nuclear accumulation of activated ERK1/2 transmitting anti-proliferation signals [61,62,63]. In our study, we demonstrated that activation of GPER by G1 induced phosphorylated ERK1/2 in Caov3 and Caov4 cells, rapidly, and ERK1/2 activation was maintained for 24 h. Compared with another study of MCF-7 cells, the peak of the p-ERK1/2 level appeared at 1 h and disappeared after 24 h during G1 treatment [14]. Whether sustained p-ERK1/2 is a bridge to the H3K4me3 accumulation and inhibition of cell growth response to activation of GPER by G1 requires further study in the future.

As we know, H3K4me3 has been generally recognized as an active promoter mark while broad H3K4me3 at the tumor suppressor genes leads to tumor suppression [25]. A previous study suggested that KDM5B, a histone demethylase, demethylated H3K4me3 to an inactive transcription state and reduced the transcription of tumor suppressor genes, then promoted gastric cancer cell proliferation and migration [28]. In addition,1α,25(OH)_2_D_3_ exerted anti-tumoral effects on breast cancer epithelial cells by increasing the level of H3K4me3 at the target gene promoters [64]. In accordance with previous studies, our study first released that GPER activation by G1 treatment had an inhibitory effect on cell growth, accompanied by elevating the level of H3K4me3 in ovarian cancer cells.

So far three main human estrogen receptors (ERα, ERβ and GPER) have been identified as potential targets for tailored endocrine therapies to treat ovarian cancer, although these approaches have not been implemented into clinical routine so far. Regarding histone modifications, previous analyses have shown that an activated endogenous estrogen-responsive TFF1 gene results in promoter recruitment of menin (MLL1/MLL2 histone methyltransferase complexes), and in the elevated trimethylation of H3K4 [65]. Moreover, the accumulation of H3K4me3 was induced on ERα-suppressed genes in the presence of ERβ, followed by epigenetic activation of transcription of tumor suppressor p53 [33]. The G1 treatment induced p53 via transcriptional and post-transcriptional pathways to suppress breast cancer cells [66]. Whether p53-dependent transcription can be activated by G1 recruitment to H3K4me3 in ovarian cancer needs further research. The relation of the GPER activation-induced p-ERK1/2 pathway with H3K4me3 in ovarian cancer need to be investigated for further therapeutic interventions or combination therapies in the future. Many substrates of ERK1/2 were found in the nucleus: they were nuclear transcriptional factors and took part in gene transcription, cell proliferation and differentiation. This is promising, because ovarian cancer cells facilitates migration and proliferation by activating the MAPK/ERK1/2 pathway [67]. On the other hand, the PI3K/AKT and ERK cascade activation induces chemotherapy resistance in endothelial ovarian cancer cells [68]. Therefore, this axis seems to be involved in the two processes that determine ovarian cancer survival most and merits further consideration.

Indirect confirmation of these results could be provided with the observation that suppression of GPER via G15 conversely promotes cell migration and proliferation of ovarian cancer cells with attenuation of p-ERK1/2 and H3K4me3 expression. The consistent results were obtained in Caov4 cells. Additionally, G15 blocked the actions of G1 partly in Caov3 cell line, in that G15 could not decrease the levels of p-ERK1/2 and H3K4me3 with promoting cell growth of Caov3 cells. The significant differences in GPER levels in Caov3 and Caov4 cells probably caused the various impacts of G15 treatment. Interestingly, in KGN cells (a human ovarian granulosa cell tumor cell line), G15 did not have an effect on cell proliferation [14]. These results might reflect the differentiation grade of granulosa tumors versus ovarian cancer subtypes at a cellular level [3]. A previous study reported that the GPER-specific antagonist G15 was able to impair GPER function and inhibit the proliferation of endometriotic cells [69]. All the observations could be explained by heterogeneity of different ovarian cell lines and different types of cancers. Our study has some limitations. One limitation is that the downstream regulators of the GPER pathway maintain phosphorylated ERK1/2 and increasing H3K4me3 levels; the localization of sustained p-ERK1/2 also was not investigated. Another limitation is that the functional cross-talk between GERP and ERα was not further studied in the present study. However, the effects of G1-induced activation of GPER on Caov3 and Caov4 cell lines are virtually identical.

## 5. Conclusions

In conclusion, our data suggests that GPER is involved in mediating the epigenetic regulation of H3K4me3 expression in ovarian cancer. Apart from the prognostic impact of the correlation of GPER and H3K4me3, we present, for the first time, that levels of phosphorylated ERK1/2 and H3K4me3 are elevated via activation of GPER by G1 treatment. Moreover, activation of GPER has an inhibitory effect on the proliferation and migration of ovarian cancer cells. Although further studies are needed, our findings are important. According to these results, the interplay of GPER and H3K4me3 could represent a biological rationale for therapeutic approaches addressing GPER as a potential target in future investigation.

## Figures and Tables

**Figure 1 cells-10-00619-f001:**
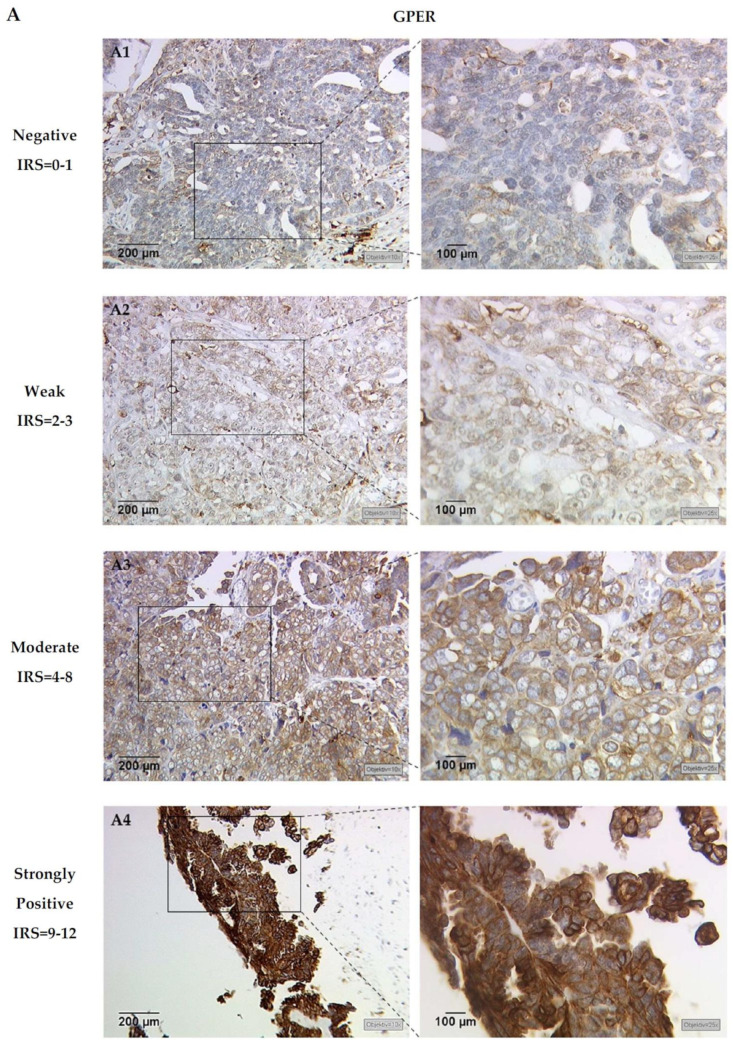
Representative microphotographs of (**A**) G-protein-coupled estrogen receptor (GPER) and (**B**) H3K4me3 expression, in the same ovarian cancer patient, are presented. A1 and B1 represent the same patient, and so forth. GPER immunohistochemical staining displays cytoplasm and membranes, while H3K4me3 shows a nucleic staining pattern in ovarian cancer specimens. Specimens were attributed to negative (IRS = 0–1, A1 and B1), weak (IRS = 2–3, A2 and B2), moderate (IRS = 4–8, A3 and B3) and strongly positive (IRS = 9–12, A4 and B4) expression levels of GPER and H3K4me3 (left panel: scale bar = 200 μm; right panel: scale bar = 100 μm).

**Figure 2 cells-10-00619-f002:**
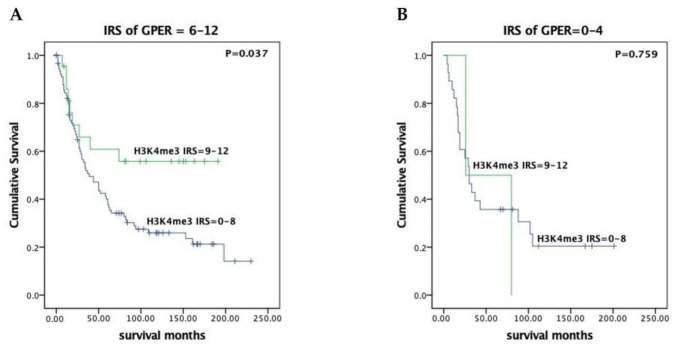
H3K4me3 associated with favorable outcome in higher GPER-expressing (IRS = 6–12) EOC patients. The prognostic significance of H3K4me3 was evaluated in subgroups of patients with high- (IRS = 6–12, (**A**)) compared to low-level (IRS = 0–4, (**B**)) GPER expression. Survival of patients with high levels of H3K4me3 expression (IRS = 9–12) (green lines) was compared to those with lower H3K4me3 expression (IRS = 0–8) (blue lines) by the log-rank text and Kaplan–Meier survival analysis. Notably, H3K4me3 predicts significantly better outcome in the subgroup classified as high-level GPER expression (left, (**A**)) compared to low-level GPER expression (right, (**B**)).

**Figure 3 cells-10-00619-f003:**
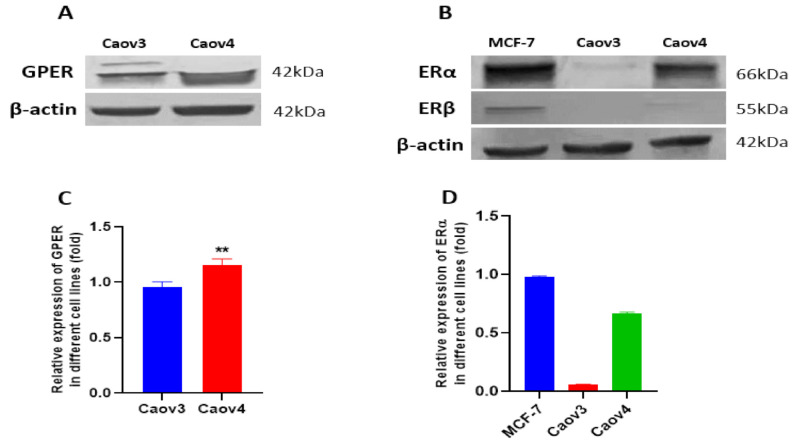
The expressions of GPER, ERα and ERβ in Caov3 and Caov4 cell lines. The protein levels of GPER and ERα were detected by Western blot analysis. (**A**) Representative example of GPER protein expression in Caov3 and Caov4 cells. (**B**) The ERα and ERβ expression in MCF-7 breast cancer cells and in Caov3 and Caov4 ovarian cancer cells. The MCF-7 cell line was seen a positive control of ERα and ERβ expressions. (**C**,**D**) Histograms represent the ratio of GPER and ERα to β-actin, respectively, as assessed with pooled densitometric data. β-actin was used as a loading control. Each experiment was repeated at least three times. The results are shown as the mean ± SEM (** *p* < 0.01).

**Figure 4 cells-10-00619-f004:**
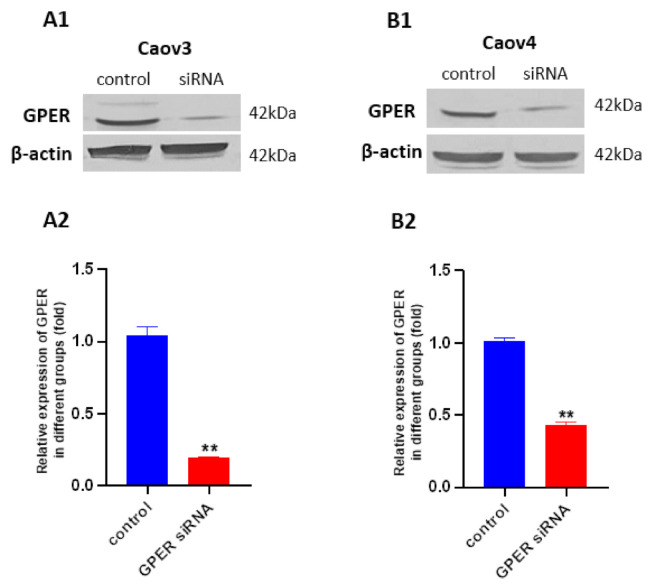
Knockdown of GPER expression with GPER siRNA in Caov3 and Caov4 cell lines. Western blot was used to analysed GPER protein levels after transfection (72 h) with or without GPER siRNA. (**A1**) Caov3; (**B1**) Caov4. Histograms compares the presence of GPER expression between control and GPER siRNA groups in Caov3 (**A2**) and Caov4 (**B2**) cell lines. Results were from one of three representative experiments and showed as mean ± SEM (** *p* < 0.01 control group vs. GPER siRNA group).

**Figure 5 cells-10-00619-f005:**
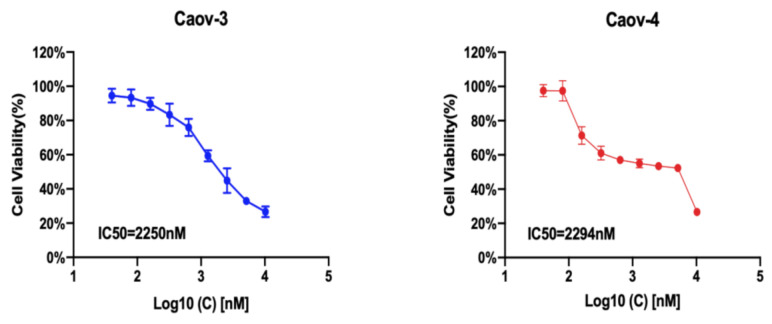
The IC50 value of G1 in different ovarian cancer cell lines. Caov3 and Caov4 cells were treated with gradient concentration of G1 for 24 h and cell viability was detected using an MTT assay. Each experiment was repeated six times (*n* = 6). The results are displayed as the means ± SD.

**Figure 6 cells-10-00619-f006:**
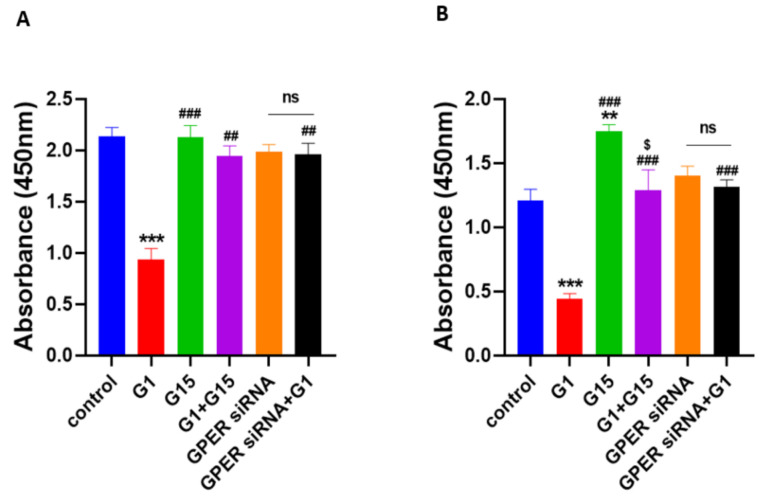
Effects of G1 and G15 treatment on cell proliferation in ovarian cancer cells. Cell proliferation was detected by the BrdU assay. Cells were treated with 1 μM G1, 1 μM G15, 1 μM G1 + 1 μM G15 and GPER-knockdown cells with the vehicle and GPER siRNA cells with 1 μM G1 for 24 h. The result was presented as effective absorbance at 450 nm. The histograms are displayed: (**A**) Caov3, and (**B**) Caov4. Six study groups in each ovarian cancer cell line were studied: control, 1 μM G1, 1 μM G15, 1 μM G1 + 1 μM G15, GPER siRNA and GPER siRNA + 1 μM G1. Each experiment was independently performed at least four times in multiple cultures. The data are represented by the mean ± SEM. Statistical analyses were performed by one-way ANOVA tests (ns *p* > 0.05, ** *p* < 0.01, *** *p* < 0.001 vs. control; ^##^
*p* < 0.01, ^###^
*p* < 0.001 vs. G1 group; ^$^
*p* < 0.05 vs. G15 group).

**Figure 7 cells-10-00619-f007:**
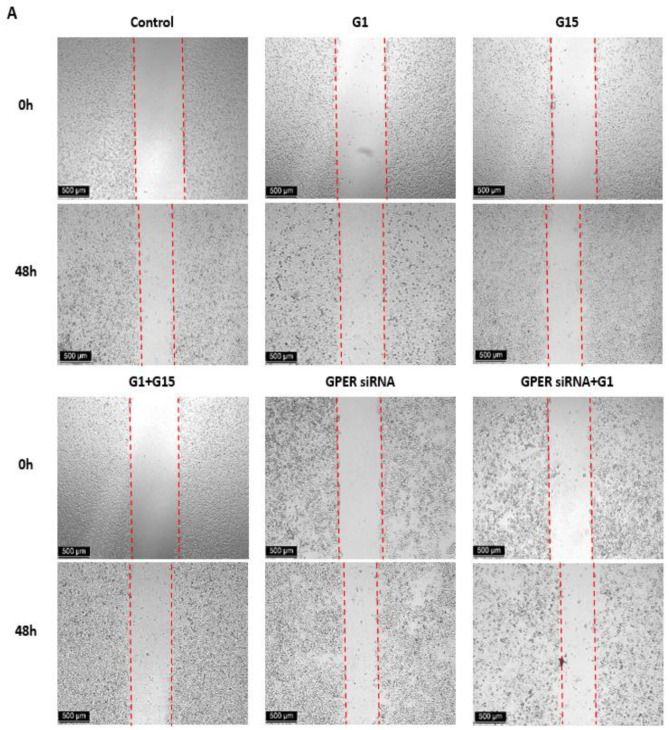
Results of wound healing scratch assay using (**A**) the Caov3 cell line and (**B**) the Caov4 cell line. The microscopy images of the wound healing assay at 0 h and 48 h. Six study groups in each ovarian cancer cell line were observed: control, 1 μM G1, 1 μM G15, 1 μM G1 + 1 μM G15, GPER siRNA group and GPER siRNA + 1 μM G1. The scale bar at the left lower corner is 500 µm. Histograms compares the presence of wound closure among the different groups in the Caov3 (**C**) and Caov4 (**D**) cell lines. Data were analyzed by ANOVA and a Tukey post-hoc test (* *p* < 0.05, ** *p* < 0.01 vs. control; ^##^
*p* < 0.01, ^###^
*p* < 0.001 vs. G1 group; ^$$^
*p* < 0.01 vs. G15 group). The results are presented as the mean ± SEM of three separate experiments (*n* = 3).

**Figure 8 cells-10-00619-f008:**
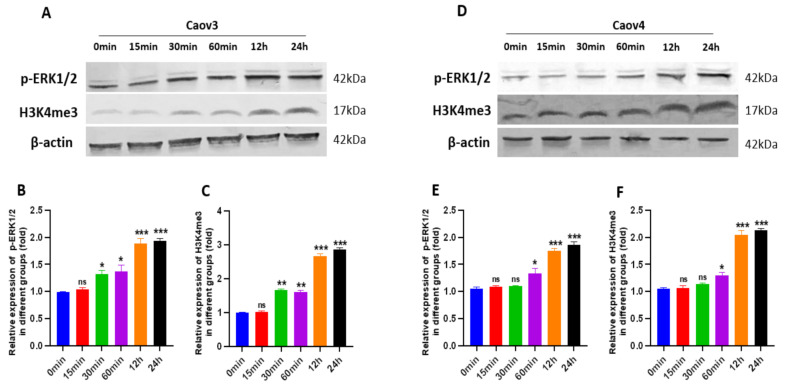
The levels of phosphorylated ERK 1/2 (p-ERK1/2) and H3K4me3 protein were shown. The levels of p-ERK1/2 and H3K4me3 were determined using Western blot analysis. (**A**–**C**) Caov3 cells were treated with 1 μM G1 for the indicated times. (**D**–**F**) Caov4 cells were treated with 1 μM G1 for the indicated times. β-actin was used as the loading control. The results are presented as the mean ± SEM of three independent experiments (*n* = 3). Data were calculated by an independent *t* test (ns *p* > 0.05, * *p* < 0.05, ** *p* < 0.01 and *** *p* < 0.001).

**Figure 9 cells-10-00619-f009:**
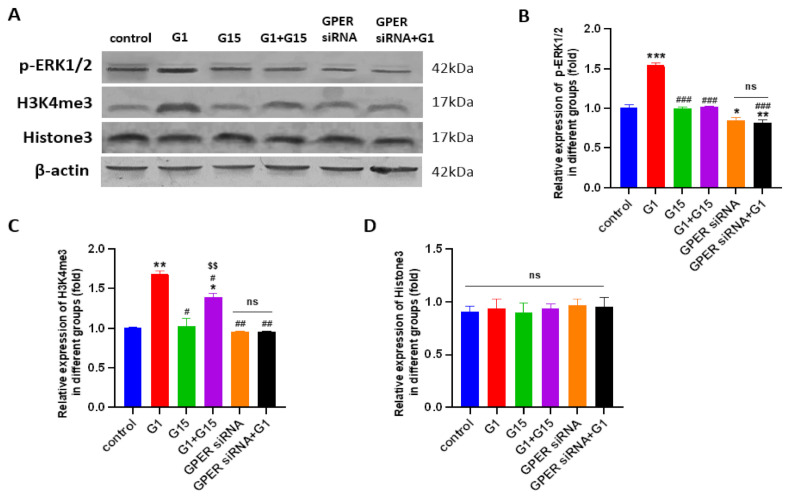
The levels of p-ERK1/2 and H3K4me3 and Histone 3 in Caov3 were detected by Western blot analysis. (**A**) The Caov3 cells were treated by 1 μM G1, 1 μM G15, 1 μM G1 + 1 μM G15, GPER siRNA with vehicle and GPER siRNA + G1 for 24 h. Histograms illustrate the ratio of (**B**) p-ERK1/2, (**C**) H3K4me3 and (**D**) Histone3. β-actin was used as the loading control. The results are presented as the mean ± SEM of three independent experiments (*n* = 3). Data were calculated by one-way ANOVA (ns *p* > 0.05, * *p* < 0.05, ** *p* < 0.01, *** *p* < 0.001 vs. control; ^#^
*p* < 0.05, ^##^
*p* < 0.01, ^###^
*p* < 0.001 vs. G1 group; ^$$^
*p* < 0.01 vs. G15 group).

**Figure 10 cells-10-00619-f010:**
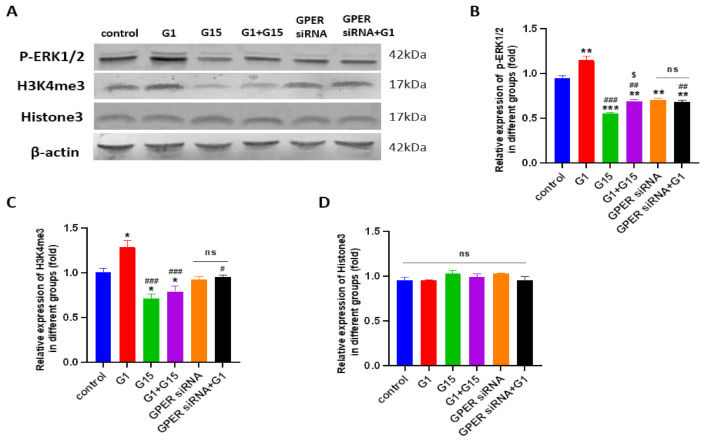
The levels of p-ERK1/2 and H3K4me3 and Histone 3 in the Caov4 cell line were detected by Western blot analysis. (**A**) The Caov4 cells were treated by 1 μM G1, 1 μM G15, 1 μM G1 + 1 μM G15, GPER siRNA with vehicle and GPER siRNA + G1 for 24 h. Histograms illustrate the ratio of (**B**) p-ERK1/2, (**C**) H3K4me3 and (**D**) Histone3. β-actin was used as the loading control. The results are presented as the mean ± SEM of three independent experiments (*n* = 3). Data were calculated by one-way ANOVA (ns *p* > 0.05, * *p* < 0.05, ** *p* < 0.01, *** *p* < 0.001 vs. control; ^#^
*p* < 0.05, ^##^
*p* < 0.01, ^###^
*p* < 0.001 vs. G1 group; ^$^
*p* < 0.05 vs. G15 group).

**Figure 11 cells-10-00619-f011:**
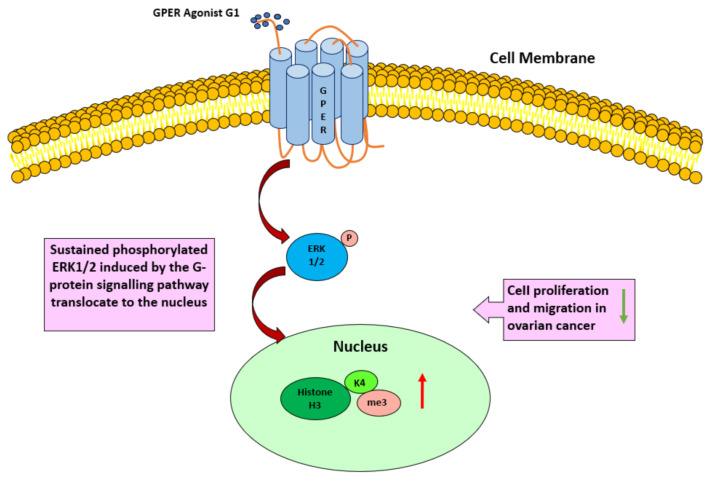
A proposed model to illustrate that activation of GPER by the selective agonist G1 regulates the level of p-ERK1/2 and H3K4me3, along with inhibition of cell proliferation and migration in ovarian cancer cells.

## Data Availability

The data presented in this study are available on request from the corresponding author. The data are not publicly available due to ethical issues.

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
