# Peer review of "The G-Protein-Coupled Estrogen Receptor (GPER) Regulates Trimethylation of Histone H3 at Lysine 4 and Represses Migration and Proliferation of Ovarian Cancer Cells In Vitro"

_cells, 2021, doi:10.3390/cells10030619_

Round 1

Reviewer 1 Report

The authors have addressed many of the concerns from the initial submission. There are only minor, but essential, editorial changes requested below:

  1. Table 1 is not necessary and can just be addressed in the text
  2. English spelling and grammar mistakes were present, minor editing is required
  3. The figure legend for Figure 3 does not include a description of subpanels C and D. Are C and D densitometry quantification of A and B, respectively? It is not clear if C and D represent mRNA or protein levels. Subpanel A seems to show significant differences in GPER expression between CAOV3 and -4 cells, but that difference isn’t reflected in the graph in panel C. Furthermore, the authors often report a more significant impact of G1/G15 treatment on Caov4 cells compared to Caov3 cells. Perhaps this difference in GPER expression levels can help explain those differences.
  4. In line 257, the authors state “In addition, ERβ was not expressed in both Caov3 and Caov4 cell lines (Figure 8B).” I think the authors meant Figure 3B, not 8B
  5. Figures S1 and S2 are described out of order. S2 is mentioned before S1 in the text, which is confusing. Likewise, Figures S3 and S4 are not mentioned anywhere in the main text.

Author Response

Reviewer 1:

The authors have addressed many of the concerns from the initial submission. There are only minor, but essential, editorial changes requested below:

1.Table 1 is not necessary and can just be addressed in the text.

-Reply

We are grateful to Reviewer’s suggestion and accept it. The manuscript has been revised, the table has deleted in the Result section (Page 5, ll.219 and Page 8, ll.229-233).

  1. English spelling and grammar mistakes were present, minor editing is required.

-Reply

We are grateful to Reviewer’s suggestion and check our manuscript again. The revised English spelling and grammar mistakes are as follows:

1). Abstract section: Page 1, ll.27; ll.31; ll.33; ll.34; ll.37

2). Introduction section:  

Page 2, ll.61-62; ll.66-68; ll.83; ll.87-88;

Page 3, ll.98 and ll.101

3). Result section:

Page 5, ll.213; Page 7, ll. 226; Page 8, ll.228; Page 9, ll.255; ll.257; ll.265

Page 10, ll.285; ll. 289; Page 11, ll.310; Page 12, ll.337; ll.338; ll.350-351; ll.354

Page 15, ll.378; ll.381-382; ll.395; ll.401-402; ll.406; Page 16, ll.412; ll.419; Page 17, ll.428

4). Discussion section:

Page 17, ll.436; ll.439; ll.445; ll.454; ll.456;

Page 18, ll.461-462; ll.471; ll.476; ll. 482; ll.495; ll.500; ll.505

Page 19, ll.521; Page 20 ll.568

5). Conclusions section: Page 21, ll.592-601.

  1. The figure legend for Figure 3 does not include a description of subpanels C and D. Are C and D densitometry quantification of A and B, respectively? It is not clear if C and D represent mRNA or protein levels. Subpanel A seems to show significant differences in GPER expression between CAOV3 and -4 cells, but that difference isn’t reflected in the graph in panel C. Furthermore, the authors often report a more significant impact of G1/G15 treatment on Caov4 cells compared to Caov3 cells. Perhaps this difference in GPER expression levels can help explain those differences.

-Reply

We thank the reviewer for suggestion and accept it. Figure 3C and 3D are the densitometry quantification of A and B respectively. The information of these two histograms was added in Figure 3 (Page 9, ll.269-270). We measured the western blot data of GPER expression in Caov3 and Caov4 cell lines again. Moreover, we repeated to calculate the result by independent two-sample t-test. The significant GPER expression was shown between Caov3 and Caov4 cell lines (Page 9 Figure 3C, ll.269 and Page 9, ll.259-260). The explanation was added in the discussion section (Page 20, ll.568-570).

  1. In line 257, the authors state “In addition, ERβ was not expressed in both Caov3 and Caov4 cell lines (Figure 8B).” I think the authors meant Figure 3B, not 8B

-Reply

We thank the reviewer for pointing out the writing mistake and revised it (Page 9, ll. 262).

5.Figures S1 and S2 are described out of order. S2 is mentioned before S1 in the text, which is confusing. Likewise, Figures S3 and S4 are not mentioned anywhere in the main text.

-Reply

We are grateful to Reviewer’s suggestion and supply the explanations of Figure S3 and S4. The Figure S3 explanation was added in the result section (Page 12, ll.339-341) and Figure S4 explanation was added in the result section (Page 15, ll.409-410).

References

  1. Chan, Q.K.Y.; Lam, H.M.; Ng, C.F.; Lee, A.Y.Y.; Chan, E.S.Y.; Ng, H.K.; Ho, S.M.; Lau, K.M. Activation of GPR30 inhibits the growth of prostate cancer cells through sustained activation of Erk1/2, c-jun/c-fos-dependent upregulation of p21, and induction of G2 cell-cycle arrest. Cell Death & Differentiation 2010, 17, 1511-1523, doi:10.1038/cdd.2010.20.
  2. Liu, Q.; Chen, Z.; Jiang, G.; Zhou, Y.; Yang, X.; Huang, H.; Liu, H.; Du, J.; Wang, H. Epigenetic down regulation of G protein-coupled estrogen receptor (GPER) functions as a tumor suppressor in colorectal cancer. Molecular Cancer 2017, 16, doi:10.1186/s12943-017-0654-3.
  3. Chen, Z.-J.; Wei, W.; Jiang, G.-M.; Liu, H.; Wei, W.-D.; Yang, X.; Wu, Y.-M.; Liu, H.; Wong, C.K.C.; Du, J., et al. Activation of GPER suppresses epithelial mesenchymal transition of triple negative breast cancer cells via NF-κB signals. Molecular Oncology 2016, 10, 775-788, doi:10.1016/j.molonc.2016.01.002.
  4. Liang, S.; Chen, Z.; Jiang, G.; Zhou, Y.; Liu, Q.; Su, Q.; Wei, W.; Du, J.; Wang, H. Activation of GPER suppresses migration and angiogenesis of triple negative breast cancer via inhibition of NF-κB/IL-6 signals. Cancer Letters 2017, 386, 12-23, doi:10.1016/j.canlet.2016.11.003.
  5. Chimento, A.; Sirianni, R.; Casaburi, I.; Zolea, F.; Rizza, P.; Avena, P.; Malivindi, R.; De Luca, A.; Campana, C.; Martire, E., et al. GPER agonist G-1 decreases adrenocortical carcinoma (ACC) cell growthin vitroandin vivo. Oncotarget 2015, 6, 19190-19203, doi:10.18632/oncotarget.4241.
  6. Vo, D.-K.H.; Hartig, R.; Weinert, S.; Haybaeck, J.; Nass, N. G-Protein-Coupled Estrogen Receptor (GPER)-Specific Agonist G1 Induces ER Stress Leading to Cell Death in MCF-7 Cells. Biomolecules 2019, 9, doi:10.3390/biom9090503.
  7. Heublein, S.; Mayr, D.; Vrekoussis, T.; Friese, K.; Hofmann, S.S.; Jeschke, U.; Lenhard, M. The G-Protein Coupled Estrogen Receptor (GPER/GPR30) is a Gonadotropin Receptor Dependent Positive Prognosticator in Ovarian Carcinoma Patients. PLoS ONE 2013, 8, doi:10.1371/journal.pone.0071791.

Reviewer 2 Report

I consider that the authors have positively answered to my questions and I recommend to accept this manuscript for publication for Cells.  

Author Response

Reviewer 2:

I consider that the authors have positively answered to my questions and I recommend to accept this manuscript for publication for Cells.  

Reply

We thank the reviewer 2 for this evaluation.

This manuscript is a resubmission of an earlier submission. The following is a list of the peer review reports and author responses from that submission.

Round 1

Reviewer 1 Report

GPER activation has been shown to regulate cancer cell growth. Both cell-specific inhibitory and stimulatory effects have been demonstrated. In ovarian cells the effects have been inhibitory. In addition GPER-mediated H3 regulation has been demonstrated.

In this manuscript the GPER agonist, G1, is reported to inhibit cell migration and growth in 2 ovarian cell lines, paralleling and increase in the ratio of H3K4me3:H3 ratios. In ovarian cancer samples they demonstrate that a subset of patients with higher expression of GPER and H3K4me3 had longer survivals

This is a reasonable study which is well-described. My concerns relate to the incomplete analysis of the correlation between GPER/H3K4me3 expression and survival as well as the characterization of GPER specific effects in the cell lines studied.

The studies on the impact of H3K4me3 expression on the prognostic impact in GPER positive EOC patients are incomplete. The rationale for the comparator groups (IRS of GPER 6-12 vs. 0-4, and IRS H3kme3 9-112 vs. 0-8) appears to be arbitrary. A more rigorous approach would be to do a 3x3 or 4x4 analysis comparing tertiles (or quartiles) of IRS for both GPER and H3K4me3 levels to analyze for trends.

Given the off target effects of both the GPER agonists and antagonists, additional confirmatory effects would better support the conclusion that the effects are GPER-dependent. This might include shRNA approaches to knock down GPER expression. Additionally, the antagonist effects should be studied in the presence of the GPER agonist, G1. (The effects of G15 seen in the absence of G1 would suggest that this was an off-target effect). Alternatively/additionally, studies using a second GPER antagonist, G36 (in the presence and absence of G1) would better support the conclusions.

Figure 8. The statistical depiction in the effects of G15 on BrdU incorporation in A2780 cells does not match the text description.

Figure 3. The G1-dependent increases in H3K4me3:H3 ratio would appear from the representative picture would appear to be more dependent on regulation of H3 expression (suggested to be used as a loading control) than H3kme3 expression. Given the apparent effect on H3 expression an alternate unrelated loading control should be utilized.

Reviewer 2 Report

This manuscript by Han et al investigates the downstream consequences of GPER activation and inactivation on epithelial ovarian cancer.  GPER is a recently discovered G-protein coupled estrogen receptor that may play an important role in gynecologic malignancies. Here, the investigators treat two ovarian cancer cells lines with a GPER agonist  (G1) and antagonist (G15) and analyze the consequences on histone H3K4me3, cancer cell migrationand proliferation. While the topic is important, this reviewer’s enthusiasm for the work is significantly decreased by the lack of controls and the potential for off-target drug effects due to the unknown GPER expression status in the cell lines chosen, the drug dosage, and the treatment time.

  1. It is unclear what the expression level of GPER is in the two chosen ovarian cancer cell lines, Caov-4 and A2780. Searching of previous literature did not reveal any previous reports of GPER in Caov-4 and only low expression in A280. The authors should show that both cell lines express GPER and that treatment with the agonist and antagonist result in altered expression of known target genes, or PI3K activation, at the doses chosen. Otherwise, it seems that the phenotypes observed in the cell culture experiments may be off-target effects, especially since most phenotypes that are observed are at higher doses of G1 after 48 hours where there appears to be apoptosis. The molecular consequences of agonists and antagonists should mostly be noticeable within a few hours of treatment. The investigators treat cells with agonist and antagonist for 48 hours, in which case many of the effects they are noticing may be a consequence of off-target effects or cell death/proliferation.

  1. The immunohistochemistry in Figures 3C and 4 is not convincing and is difficult to see, and this is further unconvincing when looking at the western blots where the change in H3K4me3 levels look essentially unchanged, by eye. When comparing immunohistochemistry to Figure 1, the intensity of staining in cells in Figures 3 and 4 look similar to the negative score. Chromatin immunoprecipitation of H3K4me3 levels at known target gene promoters would further strengthen this finding and the inclusion of positive or negative controls for comparison would further help convince the reader. In addition, zoomed-in images of single nuclei, in addition to the zoomed-out view, would make it easier to see the difference in staining. Figure 4C, top panels, look like they are different magnifications.

  1. The morphology of the nuclei at the 100nM G1 dose in Figure 3C looks dramatically different than control cells. When looking at the photographs in Figure 5, it appears that the 100nM and 1000nM concentrations of G1 may cause significant cell death, which would be a confounding factor for the wound closure assays – wound closure with G1 may be delayed only because cells are dying, not because cells are no longer migrating. This is further supported the MTT data in Figure 7. The authors, therefore, should include an analysis of apoptosis in G1 and G15 treated cells. Additionally, there is a large jump in concentrations, (i.e.: 10 to 100nM and 100 to 1000nM) and, therefore, the authors may be missing a critical dosage window where there is an effect from the drug without the confounding problem of apoptosis.

  1. Figure 8, which assesses proliferation and viability in response to G15 treatment, is concerning. The height of the bars in the graph look essentially the same (minor changes) with large error bars, so it is not convincing to this reviewer that the p value is highly significant at <0.001. An additional methodology besides a microplate-based assay is suggested to strengthen these findings, perhaps flow cytometry (of BrdU or cell cycle), cell counts over time, etc.

Reviewer 3 Report

Han et al. have proposed a manuscript entitled "G1 as selective agonist of GPER regulates H3K4me3 and represses migration and proliferation of ovarian cancer cells in vitro" for publication for Cells mdpi. As general remarks, the paper is well written with consistent results, although introduction seems to lack a literature overview. Discussion is too short. Typographical errors have also been found. However, my main concerns are of a more interpretational nature.

  1. Introduction

- Page 2, second paragraph: the functional link between GPER and ERa should be clearly addressed.

- Page 2, end of the second paragraph: A brief pharmacological profile of G-15 should also be addressed.

- Page 2, Line 50: "Estradiol", instead of estrogen, would be preferred.

- Page 2, Line 55: "Previous studies could demonstrate [...]". "Previous studies have demonstrated [...]" is more appropriate.

- Page 2, Line 59: "[...] the selective synthetic agonist [...]"

- Page 2, Line 65: "H3K4me3 [...] is closely involved in carcinogenesis [...]". This point is confusing as the authors stated in the abstract that H3K4me3 could be considered as an epigenetic signature for tumor-suppressor genes. Is H3K4me3 relevant to cell proliferation or to cell proliferation regulatory mechanisms?

- Page 2, Line 69: "[...] could be described [22].". "[...] has been described [22]." is more appropriate.

- Page 2, Line 71. Please, add appropriate references demonstrating that GPER is an estrogen receptor and note a membrane ER co-activator / regulator.

- A paragraph devoted to the significance of H4K4 me3, as a function of cell lines, must be addressed. Indeed, H4K4 me3 seems to have different meanings, depending on cell lines.

- What is exactly the role of GPER with respect to cancer. A paragraph devoted to this point is required.

  1. Results

- Page 5, Line 181: "The combination of both, GPER and H3K4me3 staining was technically feasible in 146 of 156 cases [...]". Explain why staining was not possible for 10 cases.

- Page 5, Line 181: H3K4me3 and not H3K4me.

- Page 5, Line 184: "[...] GPER staining was predominantly observed in the cytoplasm [...]". Specify the exact localization of GPER in the cytoplasm. What about localization at the plasma membrane and at the endoplasmic reticulum membrane? Both localizations have been previously evidenced.

- Page 5, Figure 1. It is difficult to analyse the microphotographs of GPER and H3K4me3. Increase the resolution. Or add a zoom in the pictures.

- Page 5, Table 1. Check.

- Page 5, Line 203: demonstrated (not demostrated). This sentence should be revised.

- Page 6, Line 220. The authors should justify the choice of the two cell lines, i.e., Caov-4 and A2780. Do they present the same phenotypes with respect two the expression of TKRs, GPER and ERs?

- Page 6, Line 223. The authors claim that the upregulation of the proliferation of the A2780 ovarian cancer cells is independent of the G1 concentration. The figure 3B seems to show a concentration-dependent process with a plateau at 100 nM of G-1. The same profile occurs with Caov-4 cells, where 10 nM seems to be inactive. Please, check.

- Page 7, Figure 3C. Increase the resolution of the Figure 3c.

- Page 7, Figure 3B. In the case of Caov-4, statistical analysis shows a significant difference (**) in the expression of H3K4me3, between the control and G1 at 10 nM. However, values seem similar. Check.

- Page 9. The Figure 5 and 6 show that G1, which is a GPER agonist, decreases the migration of ovarian cancer cells and that G-15, which is a GPER antagonist, increases the migration of cells. Similar effects have been shown with adenocortical carcinoma cells (Oncotarget, 2015, 6: 19190-19203). This article should be added in the list of references and discussed.

- To increase the impact of these results, I suggest to the authors to analyse by western blot the expression of pERK1/2 in CaOV-4 and A2780, in the presence and in the absence (control) of G-1 and G-15. These experiments should be repeated under GPER silencing conditions. What about the ERa status in these two cell lines?

- Page 11, Figure 8. The authors observe significant differences in cell growth. Check. Additionaly to high error bars, values seem similar.

  1. Discussion

- Page 11, Line 303: "High expression of both, GPER and H3K4me3, was associated with favorable prognosis." Others authors have shown that H3K4me3 was associated with late state breast cancer (Oncotarget, 2016, 7: 5094-5109). Explain such discrepancy, regarding cell lines.

- Page 11, Lines 314 to 317. These observations should be discussed in the context of the present study, where G1 and G-15, respectively, inhibits and activates cell proliferation.